# Deep direct likelihood knockoffs

**Mukund Sudarshan**[*]
Courant Institute of Mathematical Sciences
New York University
sudarshan@cims.nyu.edu

**Wesley Tansey**
Department of Epidemiology and Biostatistics
Memorial Sloan Kettering Cancer Center
tanseyw@mskcc.org

**Rajesh Ranganath**
Courant Institute of Mathematical Sciences
Center for Data Science
New York University
rajeshr@cims.nyu.edu

## Abstract

Predictive modeling often uses black box machine learning methods, such as deep neural networks, to achieve state-of-the-art performance. In scientific domains, the scientist often wishes to discover which features are actually important for making the predictions. These discoveries may lead to costly follow-up experiments and as such it is important that the error rate on discoveries is not too high. Model-X knockoffs [2] enable important features to be discovered with control of the false discovery rate (FDR). However, knockoffs require rich generative models capable of accurately modeling the knockoff features while ensuring they obey the so-called "swap" property. We develop Deep Direct Likelihood Knockoffs (DDLK), which directly minimizes the KL divergence implied by the knockoff swap property. DDLK consists of two stages: it first maximizes the explicit likelihood of the features, then minimizes the KL divergence between the joint distribution of features and knockoffs and any swap between them. To ensure that the generated knockoffs are valid under any possible swap, DDLK uses the Gumbel-Softmax trick to optimize the knockoff generator under the worst-case swap. We find DDLK has higher power than baselines while controlling the false discovery rate on a variety of synthetic and real benchmarks including a task involving a large dataset from one of the epicenters of COVID-19.

## 1 Introduction

High dimensional multivariate datasets pervade many disciplines including biology, neuroscience, and medicine. In these disciplines, a core question is which variables are important to predict the phenomenon being observed. Finite data and noisy observations make finding informative variables impossible without some error rate. Scientists therefore seek to find informative variables subject to a specific tolerance on an error rate such as the false discovery rate (FDR).

Traditional methods to control to the FDR rely on assumptions on how the covariates of interest $\mathbf{x}$ may be related to the response $\mathbf{y}$. Model-X knockoffs [2] provide an alternative framework that controls the FDR by constructing synthetic variables $\tilde{\mathbf{x}}$, called knockoffs, that look like the original covariates, but have no relationship with the response given the original covariates. Variables of this form can be used to test the conditional independence of each covariate in a collection, with the response given

---

[*]Corresponding author

the rest of the covariates by comparing the association the original covariate has with the response with the association the knockoff has with the response.

The focus on modeling the covariates shifts the testing burden to building good generative models of the covariates. Knockoffs need to satisfy two properties: (i) they need to be independent of the response given the real covariates, (ii) they need to be equal in distribution when any subset of variables is swapped between knockoffs and real data. Satisfying the first property is trivial by generating the knockoffs without looking using the response. The second property requires building conditional generative models that are able to match the distribution of the covariates.

**Related work.** Existing knockoff generation methods can be broadly classified as either model-specific or flexible. Model-specific methods such as hidden markov models (HMMs) [22] or Auto-Encoding Knockoffs [15] make assumptions about the covariate distribution, which can be problematic if the data does not satisfy these assumptions. HMMs assume the joint distribution of the covariates can be factorized into a markov chain. Auto-Encoding Knockoffs use variational auto-encoders (VAEs) to model $\mathbf{x}$ and sample knockoffs $\widetilde{\mathbf{x}}$. VAEs assume $\mathbf{x}$ lies near a low dimensional manifold, whose dimension is controlled by a latent variable. Covariates that violate this low-dimensional assumption can be better modeled by increasing the dimension of the latent variable, but risk retaining more information about $\mathbf{x}$, which can reduce the power to select important variables.

Flexible methods for generating knockoffs such as KnockoffGAN [10] or Deep Knockoffs [20] focus on likelihood-free generative models. KnockoffGAN uses generative adversarial network (GAN)-based generative models, which can be difficult to estimate [18] and sensitive to hyperparameters [21, 8, 17]. Deep Knockoffs employ maximum mean discrepancys (MMDs), the effectiveness of which often depends on the choice of a kernel which can involve selecting a bandwidth hyperparameter. Ramdas et al. [19] show that in several cases, across many choices of bandwidth, MMD approaches 0 as dimensionality increases while KL divergence remains non-zero, suggesting MMDs may not reliably generate high-dimensional knockoffs. Deep Knockoffs also prevent the knockoff generator from memorizing the covariates by explicitly controlling the correlation between the knockoffs and covariates. This is specific to second order moments, and may ignore higher order ones present in the data.

We propose deep direct likelihood knockoffs (DDLK), a likelihood-based method for generating knockoffs without the use of latent variables. DDLK is a two stage algorithm. The first stage uses maximum likelihood to estimate the distribution of the covariates from observed data. The second stage estimates the knockoff distribution with likelihoods by minimizing the Kullback–Leibler divergence (KL) between the joint distribution of the real covariates with knockoffs and the joint distribution of any swap of coordinates between covariates and knockoffs. DDLK expresses the likelihoods for swaps in terms of the original joint distribution with real covariates swapped with knockoffs. Through the Gumbel-Softmax trick [9, 16], we optimize the knockoff distribution under the worst swaps. By ensuring that the knockoffs are valid in the worst cases, DDLK learns valid knockoffs in all cases. To prevent DDLK from memorizing covariates, we introduce a regularizer to encourage high conditional entropy for the knockoffs given the covariates. We study DDLK on synthetic, semi-synthetic, and real datasets. Across each study, DDLK controls the FDR while achieving higher power than competing GAN-based, MMD-based, and autoencoder-based methods.

## 2 Knockoff filters

Model-X knockoffs [2] is a tool used to build variable selection methods. Specifically, it facilitates the control of the FDR, which is the proportion of selected variables that are not important. In this section, we review the requirements to build variable selection methods using Model-X knockoffs.

Consider a data generating distribution $q(\mathbf{x})q(\mathbf{y} \mid \mathbf{x})$ where variables $\mathbf{x} \in \mathbb{R}^d$, response $\mathbf{y}$ only depends on $\mathbf{x}_S$, $S \subseteq [d]$, and $\mathbf{x}_S$ is a subset of the variables. Let $\hat{S}$ be a set of indices identified by a variable selection algorithm. The goal of such algorithms is to return an $\hat{S}$ that maximizes the number of indices in $S$ while maintaining the FDR at some nominal level:

$$\text{FDR} = \mathbb{E}\left[ \frac{|\{j : j \in \hat{S} \setminus S\}|}{\max\left(|\{j : j \in \hat{S}\}|, 1\right)} \right].$$

To control the FDR at the nominal level, Model-X knockoffs requires (a) knockoffs $\widetilde{\mathbf{x}}$, and (b) a knockoff statistic $w_j$ to assess the importance of each feature $\mathbf{x}_j$.

Knockoffs $\widetilde{\mathbf{x}}$ are random vectors that satisfy the following properties for any set of indices $H \subseteq [d]$:

$$[\mathbf{x}, \widetilde{\mathbf{x}}] \overset{d}{=} [\mathbf{x}, \widetilde{\mathbf{x}}]_{\text{swap}(H)} \tag{1}$$

$$\mathbf{y} \perp \widetilde{\mathbf{x}} \mid \mathbf{x}. \tag{2}$$

The swap property eq. (1) ensures that the joint distribution of $[\mathbf{x}, \widetilde{\mathbf{x}}]$ is invariant under any swap. A swap operation at position $j$ is defined as exchanging the entry of $\mathbf{x}_j$ and $\widetilde{\mathbf{x}}_j$. For example, when $\mathbf{x} = [\mathbf{x}_1, \mathbf{x}_2, \mathbf{x}_3]$, and $H = \{1, 3\}$, $[\mathbf{x}, \widetilde{\mathbf{x}}]_{\text{swap}(H)} = [\widetilde{\mathbf{x}}_1, \mathbf{x}_2, \widetilde{\mathbf{x}}_3, \mathbf{x}_1, \widetilde{\mathbf{x}}_2, \mathbf{x}_3]$. Equation (2) ensures that the response $\mathbf{y}$ is independent of the knockoff $\widetilde{\mathbf{x}}$ given the original features $\mathbf{x}$.

A knockoff statistic $w_j$ must satisfy the flip-sign property. This means if $\mathbf{x}_j \in S$, $w_j$ must be positive. Otherwise, the sign of $w_j$ must be positive or negative with equal probability.

Given knockoff $\widetilde{\mathbf{x}}$ and knockoff statistics $\{w_j\}_{j=1}^d$, exact control of the FDR at level $p$ can be obtained by selecting variables in $\mathbf{x}$ such that $w_j > \tau_p$. The threshold $\tau_p$ is given by:

$$\tau_p = \min_{t>0} \left\{ t : \frac{1 + |\{j : w_j \leq -t\}|}{|\{j : w_j \geq t\}|} \leq p \right\}. \tag{3}$$

While knockoffs are a powerful tool to ensure that the FDR is controlled at the nominal level, the choice of method to generate knockoffs is left to the practitioner.

Existing methods for knockoffs include model-specific approaches that make specific assumptions about the covariate distribution, and flexible likelihood-free methods. If the joint distribution of $\mathbf{x}$ cannot be factorized into a markov chain [22] or if $\mathbf{x}$ does not lie near a low-dimensional manifold [15], model-specific generators will yield knockoffs that are not guaranteed to control the FDR. Likelihood-free generation methods that use GANs [10] or MMDs [20] make fewer assumptions about $\widetilde{\mathbf{x}}$, but can be difficult to estimate [18], sensitive to hyperparameters [21, 8, 17], or suffer from low power in high dimensions [19]. In realistic datasets, where $\mathbf{x}$ can come from an arbitrary distribution and dimensionality is high, it remains to be seen how to reliably generate knockoffs that satisfy eqs. (1) and (2).

## 3 Deep direct likelihood knockoffs

We motivate DDLK with the following observation. The swap property in eq. (1) is satisfied if the KL divergence between the original and swapped distributions is zero. Formally, let $H$ be a set of indices to swap, and $\mathbf{z} = [\mathbf{x}, \widetilde{\mathbf{x}}]$, $\mathbf{w} = [\mathbf{x}, \widetilde{\mathbf{x}}]_{\text{swap}(H)}$. Then under any such $H \subseteq [d]$:

$$\text{KL}(q_{\mathbf{z}} \parallel q_{\mathbf{w}}) = \mathbb{E}_{q_{\mathbf{z}}(\mathbf{z})} \left[ \log \frac{q_{\mathbf{z}}(\mathbf{z})}{q_{\mathbf{w}}(\mathbf{z})} \right] = 0. \tag{4}$$

A natural algorithm for generating valid knockoffs might be to parameterize each distribution above and solve for the parameters by minimizing the LHS of eq. (4). However, modeling $q_{\mathbf{w}}$ for every possible swap is difficult and computationally infeasible in high dimensions. Theorem 3.1 provides a useful solution to this problem.

**Theorem 3.1.** *Let $\mu$ be a probability measure defined on a measurable space. Let $f_H$ be a swap function using indices $H \subseteq [d]$. If $\mathbf{v}$ is a sample from $\mu$, the probability law of $f_H(\mathbf{v})$ is $\mu \circ f_H$.*

As an example, in the continuous case, where $q_{\mathbf{z}}$ and $q_{\mathbf{w}}$ are the densities of $\mathbf{z}$ and $\mathbf{w}$ respectively, $q_{\mathbf{w}}$ evaluated at a sample $\mathbf{v}$ is simply $q_{\mathbf{z}}$ evaluated at the swap of $\mathbf{v}$. We show the direct proof of this example and theorem 3.1 in appendix A. A useful consequence of theorem 3.1 is that DDLK needs to only model $q_{\mathbf{z}}$, instead of $q_{\mathbf{z}}$ and every possible swap distribution $q_{\mathbf{w}}$. To derive the DDLK algorithm, we first expand eq. (4):

$$\mathbb{E}_{q_{\mathbf{z}}(\mathbf{z})} \left[ \log \frac{q_{\mathbf{z}}(\mathbf{z})}{q_{\mathbf{w}}(\mathbf{z})} \right] = \mathbb{E}_{q(\mathbf{x})} \mathbb{E}_{q(\widetilde{\mathbf{x}}|\mathbf{x})} \left[ \log \frac{q(\mathbf{x})q(\widetilde{\mathbf{x}} \mid \mathbf{x})}{q(\mathbf{u})q(\widetilde{\mathbf{u}} \mid \mathbf{u})} \right], \tag{5}$$

where $[\mathbf{u}, \widetilde{\mathbf{u}}] = [\mathbf{x}, \widetilde{\mathbf{x}}]_{\text{swap}(H)}$. DDLK models the RHS by parameterizing $q(\mathbf{x})$ and $q(\widetilde{\mathbf{x}} \mid \mathbf{x})$ with $\hat{q}_{\text{joint}}(\mathbf{x}; \theta)$ and $\hat{q}_{\text{knockoff}}(\widetilde{\mathbf{x}} \mid \mathbf{x}; \phi)$ respectively. The parameters $\theta$ and $\phi$ can be optimized separately in two stages.

**Algorithm 1** DDLK

---

**Input:** $\mathcal{D}_N := \{\boldsymbol{x}^{(i)}\}_{i=1}^N$, dataset of covariates; $\lambda$, regularization parameter; $\alpha_\phi$, learning rate for $\hat{q}_{\text{knockoff}}$; $\alpha_\beta$, learning rate for $\hat{q}_{\text{gumbel}}$

**Output:** $\theta$, parameter for $\hat{q}_{\text{joint}}$, $\hat{q}_{\text{knockoff}}$, parameter for $\hat{q}_{\text{knockoff}}$

$\theta = \arg\max_\theta \frac{1}{N} \sum_{i=1}^N \log \hat{q}_{\text{joint}}(\boldsymbol{x}^{(i)}; \theta)$

**while** $\hat{q}_{knockoff}$ *not converged* **do**

    Sample $\{\widetilde{\boldsymbol{x}}^{(i)}\}_{i=1}^N$, where $\widetilde{\boldsymbol{x}}^{(i)} \sim \hat{q}_{\text{knockoff}}(\widetilde{\mathbf{x}} \mid \boldsymbol{x}^{(i)}; \phi)$

    Sample swap $H \sim \hat{q}_{\text{gumbel}}(\beta)$

    Create $\{(\boldsymbol{u}^{(i)}, \widetilde{\boldsymbol{u}}^{(i)})\}_{i=1}^N$, where $[\boldsymbol{u}^{(i)}, \widetilde{\boldsymbol{u}}^{(i)}] = [\boldsymbol{x}^{(i)}, \widetilde{\boldsymbol{x}}^{(i)}]_{\text{swap}(H)}$

    Let $\mathcal{A}(\phi) = \frac{1}{N} \sum_{i=1}^N \log \hat{q}_{\text{joint}}(\boldsymbol{x}^{(i)}; \theta) + (1 + \lambda) \log \hat{q}_{\text{knockoff}}(\widetilde{\boldsymbol{x}}^{(i)} \mid \boldsymbol{x}^{(i)}; \phi)$

    Let $\mathcal{B}(\phi, \beta) = \frac{1}{N} \sum_{i=1}^N \log \hat{q}_{\text{joint}}(\boldsymbol{u}^{(i)}; \theta) + \log \hat{q}_{\text{knockoff}}(\widetilde{\boldsymbol{u}}^{(i)} \mid \boldsymbol{u}^{(i)}; \phi)$

    $\phi \leftarrow \phi - \alpha_\phi \nabla_\phi(\mathcal{A}(\phi) - \mathcal{B}(\phi, \beta))$

    $\beta \leftarrow \beta + \alpha_\beta \nabla_\beta(\mathcal{A}(\phi) - \mathcal{B}(\phi, \beta))$

**end**

**return** $\theta$, $\phi$, $\beta$

---

**Stage 1: Covariate distribution estimation.** We model the distribution of $\mathbf{x}$ using $\hat{q}_{\text{joint}}(\mathbf{x}; \theta)$. The parameters of the model $\theta$ are learned by maximizing $\mathbb{E}_{\boldsymbol{x} \sim \mathcal{D}_N}[\log \hat{q}_{\text{joint}}(\boldsymbol{x}; \theta)]$ over a dataset $\mathcal{D}_N := \{\boldsymbol{x}^{(i)}\}_{i=1}^N$ of $N$ samples.

**Stage 2: Knockoff generation.** For any fixed swap $H$, minimizing the KL divergence between the following distributions ensures the swap property eq. (1) required of knockoffs:

$$\text{KL}(\hat{q}_{\text{joint}}(\mathbf{x}; \theta)\hat{q}_{\text{knockoff}}(\widetilde{\mathbf{x}} \mid \mathbf{x}; \phi) \,\|\, \hat{q}_{\text{joint}}(\mathbf{u}; \theta)\hat{q}_{\text{knockoff}}(\widetilde{\mathbf{u}} \mid \mathbf{u}; \phi)). \tag{6}$$

Fitting the knockoff generator $\hat{q}_{\text{knockoff}}(\widetilde{\mathbf{x}} \mid \mathbf{x}; \phi)$ involves minimizing this KL divergence for all possible swaps $H$. To make this problem tractable, we use several building blocks that help us (a) sample swaps with the highest values of this KL and (b) prevent $\hat{q}_{\text{knockoff}}$ from memorizing $\mathbf{x}$ to trivially satisfy the swap property in eq. (1).

### 3.1 Fitting DDLK

Knockoffs must satisfy the swap property eq. (1) for all potential sets of swap indices $H \subseteq [d]$. While this seems to imply that the KL objective in eq. (6) must be minimized under an exponential number of swaps, swapping every coordinate suffices [3]. More generally, showing the swap property for a collection of sets where every coordinate can be represented as the symmetric difference of members of the collection is sufficient. See appendix A.4 more more details.

**Sampling swaps.** Swapping $d$ coordinates can be expensive in high dimensions, so existing methods resort to randomly sampling swaps [20, 10] during optimization. Rather than sample each coordinate uniformly at random, we propose parameterizing the sampling process for swap indices $H$ so that swaps sampled from this process yields large values of the KL objective in eq. (6). We do so because of the following property of swaps, which we prove in appendix A.

**Lemma 3.2.** *Worst case swaps: Let $q(H; \beta)$ be the worst case swap distribution. That is, the distribution over swap indices that maximizes*

$$\mathbb{E}_{H \sim q(H;\beta)} KL(\hat{q}_{joint}(\mathbf{x}; \theta)\hat{q}_{knockoff}(\widetilde{\mathbf{x}} \mid \mathbf{x}; \phi) \,\|\, \hat{q}_{joint}(\mathbf{u}; \theta)\hat{q}_{knockoff}(\widetilde{\mathbf{u}} \mid \mathbf{u}; \phi)) \tag{7}$$

*with respect to $\beta$. If eq. (7) is minimized with respect to $\phi$, knockoffs sampled from $\hat{q}_{knockoff}$ will satisfy the swap property in eq. (1) for any swap $H$ in the power set of $[d]$.*

Randomly sampling swaps can be thought of as sampling from $d$ Bernoulli random variables $\{\mathbf{b}_j\}_{j=1}^d$ with parameters $\beta = \{\beta_j\}_{j=1}^d$ respectively, where each $\mathbf{b}_j$ indicates whether the $j$th coordinate is to be swapped. A set of indices $H$ can be generated by letting $H = \{j : \mathbf{b}_j = 1\}$. To learn a sampling process that helps maximize eq. (7), we optimize the values of $\beta$. However, since score function gradients for the parameters of Bernoulli random variables can have high variance, DDLK uses a continuous relaxation instead. For each coordinate $j \in d$, DDLK learns the parameters for a Gumbel-Softmax [9, 16] distribution $\hat{q}_{\text{gumbel}}(\beta_j)$.

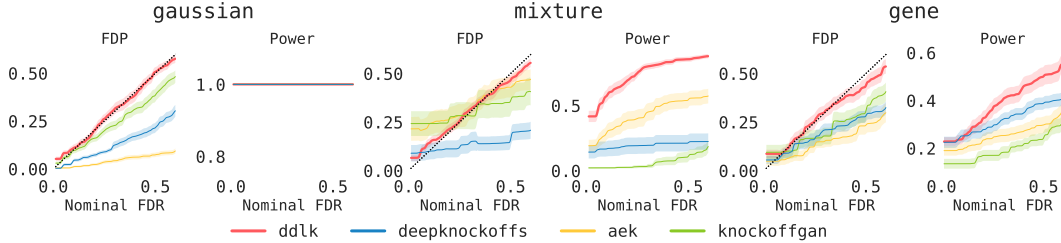

**Figure 1: DDLK controls FDR at the nominal rate and achieves highest power on a variety of benchmarks.** For each benchmark, we show the false discovery proportion (FDP) and power of each knockoff method.

**Entropy regularization.** Minimizing the the KL objective in eq. (6) over the worst case swap distribution will generate knockoffs that satisfy the swap property eq. (1). However, a potential solution in the optimization of $\hat{q}_{\text{knockoff}}(\widetilde{\mathbf{x}} \mid \mathbf{x})$ is to memorize the covariates $\mathbf{x}$, which reduces the power to select important variables.

To solve this problem, DDLK introduces a regularizer based on the conditional entropy, to push $\widetilde{\mathbf{x}}$ to not be a copy of $\mathbf{x}$. This regularizer takes the form $-\lambda \mathbb{E}[-\log \hat{q}_{\text{knockoff}}(\widetilde{\mathbf{x}} \mid \mathbf{x}; \phi)]$, where $\lambda$ is a hyperparameter.

Including the regularizer on conditional entropy, and Gumbel-Softmax sampling of swap indices, the final optimization objective for DDLK is:

$$\min_{\phi} \max_{\beta} \mathbb{E}_{H \sim \hat{q}_{\text{gumbel}}(\beta)} \mathbb{E}_{\boldsymbol{x} \sim \mathcal{D}_N} \mathbb{E}_{\widetilde{\boldsymbol{x}} \sim \hat{q}_{\text{knockoff}}(\widetilde{\mathbf{x}}|\boldsymbol{x};\phi)} \log \frac{\hat{q}_{\text{joint}}(\boldsymbol{x};\theta)\hat{q}_{\text{knockoff}}(\widetilde{\boldsymbol{x}} \mid \boldsymbol{x};\phi)^{1+\lambda}}{\hat{q}_{\text{joint}}(\boldsymbol{u};\theta)\hat{q}_{\text{knockoff}}(\widetilde{\boldsymbol{u}} \mid \boldsymbol{u};\phi)} \quad (8)$$

where $[\boldsymbol{u}, \widetilde{\boldsymbol{u}}] = [\boldsymbol{x}, \widetilde{\boldsymbol{x}}]_{\text{swap}(H)}$. We show the full DDLK algorithm in algorithm 1. DDLK fits $\hat{q}_{\text{joint}}$ by maximizing the likelihood of the data. It then fits $\hat{q}_{\text{knockoff}}$ by optimizing eq. (8) with noisy gradients. To do this, DDLK first samples knockoffs conditioned on the covariates and a set of swap coordinates, then computes Monte-Carlo gradients of the DDLK objective in eq. (8) with respect to parameters $\phi$ and $\beta$. In practice DDLK can use stochastic gradient estimates like the score function or reparameterization gradients for this step. The $\hat{q}_{\text{joint}}$ and $\hat{q}_{\text{knockoff}}$ models can be implemented with flexible models like MADE [7] or mixture density networks [1].

## 4 Experiments

We study the performance of DDLK on several synthetic, semi-synthetic, and real-world datasets. We compare DDLK with several non-Gaussian knockoff generation methods: Auto-Encoding Knockoffs (AEK) [15], KnockoffGAN [10], and Deep Knockoffs [20].[2]

Each experiment involves three stages. First, we fit a knockoff generator using a dataset of covariates $\{\boldsymbol{x}^{(i)}\}_{i=1}^{N}$. Next, we fit a response model $\hat{q}_{\text{response}}(\mathbf{y} \mid \mathbf{x}; \gamma)$, and use its performance on a held-out set to create a knockoff statistic $w_j$ for each feature $\mathbf{x}_j$. We detail the construction of these test statistics in appendix E. Finally, we apply a knockoff filter to the statistics $\{w_j\}_{j=1}^{d}$ to select features at a nominal FDR level $p$, and measure the ability of each knockoff method to select relevant features while maintaining FDR at $p$. For the synthetic and semi-synthetic tasks, we repeat these three stages 30 times to obtain interval estimates of each performance metric.

In our experiments, we assume $\mathbf{x}$ to be real-valued with continuous support and decompose the models $\hat{q}_{\text{joint}}$ and $\hat{q}_{\text{knockoff}}$ via the chain rule: $q(\mathbf{x} \mid \cdot) = q(\mathbf{x}_1 \mid \cdot) \prod_{j=2}^{d} q(\mathbf{x}_j \mid \cdot, \mathbf{x}_1, \ldots, \mathbf{x}_{j-1})$. For each conditional distribution, we fit a mixture density network [1] where the number of mixture components is a hyperparameter. In principle, any model that produces an explicit likelihood value can be used to model each conditional distribution.

Fitting $\hat{q}_{\text{joint}}$ involves using samples from a dataset, but fitting $\hat{q}_{\text{knockoff}}$ involves sampling from $\hat{q}_{\text{knockoff}}$. This is a potential issue because the gradient of the DDLK objective eq. (8) with respect to $\phi$ is

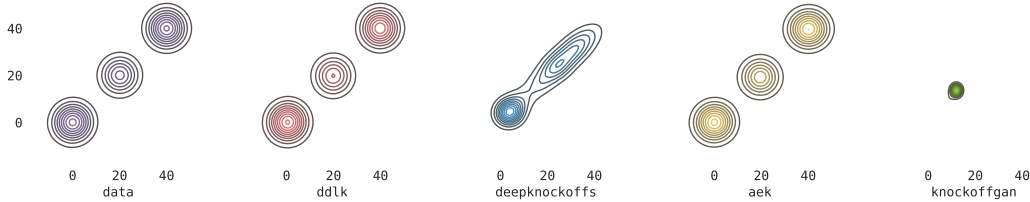

**Figure 2: DDLK closely models the modes, covariances, and mixture proportions of the `mixture` dataset**. Auto-Encoding Knockoffs also capture every mode, but does so by overfitting to the covariates. Deep Knockoffs are able to match the first two moments, but fail to capture every mode. KnockoffGAN suffers from mode collapse and fails to capture every mode.

difficult to compute as it involves integrating $\hat{q}_{\text{knockoff}}$, which depends on $\phi$. To solve this, we use an implicit reparameterization [5] of mixture densities. Further details of this formulation are presented in appendix B.

Across each benchmark involving DDLK, we vary only the $\lambda$ entropy regularization parameter based on the amount of dependence among covariates. The number of parameters, learning rate, and all other hyperparameters are kept constant. To sample swaps $H$, we sample using a straight-through Gumbel-Softmax estimator [9]. This allows us to sample binary values for each swap, but use gradients of a continuous approximation during optimization. For brevity, we present the exact hyperparameter details for DDLK in appendix C.

We run each experiment on a single CPU with 4GB of memory. DDLK takes roughly 40 minutes in total to fit both $\hat{q}_{\text{joint}}$ and $\hat{q}_{\text{knockoff}}$ on a 100-dimensional dataset.

**Synthetic benchmarks.** Our tests on synthetic data seek to highlight differences in power and FDR between each knockoff generation method. Each dataset in this section consists of $N = 2000$ samples, 100 features, 20 of which are used to generate the response $\mathbf{y}$. Testing the global null (0 important features) can also help understand the performance of a knockoff method, as FDR control equates to control of the family-wise error rate: a stricter notion of false discovery control. However, we found our results in global null experiments to be no more instructive about the differences between each method than with 20 important features. We split the data into a training set (70%) to fit each knockoff method, a validation set (15%) used to tune the hyperparameters of each method, and a test set (15%) for evaluating knockoff statistics.

[`gaussian`]: We first replicate the multivariate normal benchmark of Romano et al. [20]. We sample $\mathbf{x} \sim \mathcal{N}(0, \Sigma)$, where $\Sigma$ is a $d$-dimensional covariance matrix whose entries $\Sigma_{i,j} = \rho^{|i-j|}$. This autoregressive Gaussian data exhibits strong correlations between adjacent features, and lower correlations between features that are further apart. We generate $\mathbf{y} \mid \mathbf{x} \sim \mathcal{N}(\langle \mathbf{x}, \alpha \rangle, 1)$, where coefficients for the important features are drawn as $\alpha_j \sim \frac{100}{\sqrt{N}} \cdot \text{Rademacher}(0.5)$. In our experiments, we set $\rho = 0.6$. We let the DDLK entropy regularization parameter $\lambda = 0.1$. Our model $\hat{q}_{\text{response}}$ for $\mathbf{y} \mid \mathbf{x}$ is a 1-layer neural network with 200 parameters.

[`mixture`]: To compare each method on its ability to generate non-Gaussian knockoffs, we use a mixture of autoregressive Gaussians. This is a more challenging benchmark as each covariate is multi-modal, and highly correlated with others. We sample $\mathbf{x} \sim \sum_{k=1}^{K} \pi_k \mathcal{N}(\mu_k, \Sigma_k)$, where each $\Sigma_k$ is a $d$-dimensional covariance matrix whose $(i, j)$th entry is $\rho_k^{|i-j|}$. We generate $\mathbf{y} \mid \mathbf{x} \sim \mathcal{N}(\langle \mathbf{x}, \alpha \rangle, 1)$, where coefficients for the important features are drawn as $\alpha_j \sim \frac{100}{\sqrt{N}} \cdot \text{Rademacher}(0.5)$. In our experiments, we set $K = 3$, and $(\rho_1, \rho_2, \rho_3) = (0.6, 0.4, 0.2)$. Cluster centers are set to $(\mu_1, \mu_2, \mu_3) = (0, 20, 40)$, and mixture proportions are set to $(\pi_1, \pi_2, \pi_3) = (0.4, 0.2, 0.4)$. We let the DDLK entropy regularization parameter $\lambda = 0.001$. Figure 2 visualizes two randomly selected dimensions of this data.

***Results.*** Figure 1 compares the average FDP and power (percentage of important features selected) of each knockoff generating method. The average FDP is an empirical estimate of the FDR. In the case of the `gaussian` dataset, all methods control FDR at or below the the nominal level, while achieving 100% power to select important features. The main difference between each method is in the calibration of null statistics. Recall that a knockoff filter assumes a null statistic to be positive or

negative with equal probability, and features with negative statistics below a threshold are used to control the number of false discoveries when features with positive statistics above the same threshold are selected. DDLK produces the most well calibrated null statistics as evidenced by the closeness of its FDP curve to the dotted diagonal line.

Figure 1 also demonstrates the effectiveness of DDLK in modeling non-Gaussian covariates. In the case of the `mixture` dataset, DDLK achieves significantly higher power than the baseline methods, while controlling the FDR at nominal levels. To understand why this may be the case, we plot the joint distribution of two randomly selected features in fig. 2. DDLK and Auto-Encoding Knockoffs both seem to capture all three modes in the data. However, Auto-Encoding Knockoffs tend to produce knockoffs that are very similar to the original features, and yield lower power when selecting variables, shown in fig. 1. Deep Knockoffs manage to capture the first two moments of the data – likely due to an explicit second-order term in the objective function – but tend to over-smooth and fail to properly estimate the knockoff distribution. KnockoffGAN suffers from mode collapse, and fails to capture even the first two moments of the data. This yields knockoffs that not only have low power, but also fail to control FDR at nominal levels.

**Robustness of DDLK to entropy regularization.**     To provide guidance on how to set the entropy regularization parameter, we explore the effect of $\lambda$ on both FDR control and power. Intuitively, lower values of $\lambda$ will yield solutions of $\hat{q}_{\text{knockoff}}$ that may satisfy eq. (1) and control FDR well, but may also memorize the covariates and yield low power. Higher values of $\lambda$ may help improve power, but at the cost of FDR control. In this experiment, we again use the `gaussian` dataset, but vary $\lambda$ and the correlation parameter $\rho$. Figure 3 highlights the performance of DDLK over various settings of $\lambda$ and $\rho$. We show a heatmap where each cell represents the RMSE between the nominal FDR and mean FDP curves over 30 simulations. In each of these settings DDLK achieves a power of 1, so we only visualize FDP. We observe that the FDP of DDLK is very close to its expected value for most settings where $\lambda \leq 0.1$. This is true over a wide range of $\rho$ explored, demonstrating that DDLK is not very sensitive to the choice of this hyperparameter. We also notice that data with weaker correlations see a smaller increase in FDP with larger values of $\lambda$. In general, checking the FDP on synthetic responses generated conditional on real covariates can aid in selecting $\lambda$.

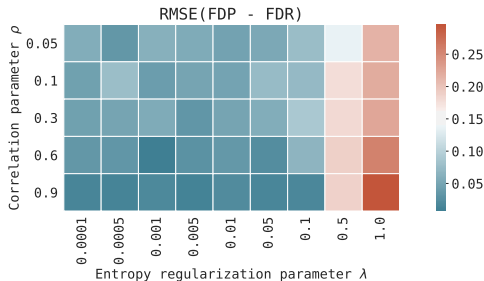

**Figure 3:** **DDLK is robust to choices of entropy regularization parameter** $\lambda$. For most choices of $\lambda \leq 0.1$, DDLK achieves FDP very close to that of the nominal FDR rate. This figure shows the RMSE between the expected and actual FDP curves.

**Semi-synthetic benchmark.**     [`gene`]: In order to evaluate the FDR and power of each knockoff method using covariates found in a genomics context, we create a semi-synthetic dataset. We use RNA expression data of 963 cancer cell lines from the Genomics of Drug Sensitivity in Cancer study [24]. Each cell line has expression levels for 20K genes, of which we sample 100 such that every feature is highly correlated with at least one other feature. We create 30 independent replications of this experiment by repeating the following process. We first sample a gene $\mathbf{x}_1$ uniformly at random, adding it to the set $\mathcal{X}$. For $\mathbf{x}_j, j > 1$, we sample $\mathbf{x}_k$ uniformly at random from $\mathcal{X}$ and compute the set of 50 genes not in $\mathcal{X}$ with the highest correlation with $\mathbf{x}_k$. From this set of 50, we uniformly sample a gene $\mathbf{x}_j$ and add it to the feature set. We repeat this process for $j = 2, \ldots, 100$, yielding 100 genes in total.

We generate $\mathbf{y} \mid \mathbf{x}$ using a nonlinear response function adapted from a study on feature selection in neural network models of gene-drug interactions [14]. The response consists of a nonlinear term, a second-order term, and a linear term. For brevity, appendix F contains the full simulation details. We let DDLK entropy regularization parameter $\lambda = 0.001$.

***Results.***     Figure 1 (right) highlights the empirical FDP and power of each knockoff generating method in the context of `gene`. All methods control the FDR below the nominal level, but the average FDP of DDLK at FDR thresholds below 0.3 is closer to its expected value. This range of thresholds is especially important as nominal levels of FDR below 0.3 are most used in practice. In this range, DDLK achieves power on par with Deep Knockoffs at levels below 0.1, and higher power

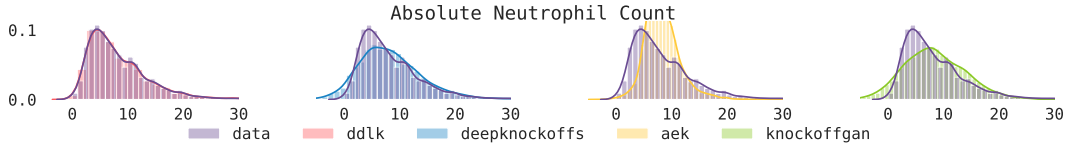

**Figure 4: DDLK learns the marginals of COVID-19 data better than competing baselines.** We plot the marginal distribution of a feature in a COVID-19 dataset, and the corresponding marginal of samples from each knockoff method.

| Feature | **DDLK** | Deep Knockoffs | AEK | KnockoffGAN | Validated |
|---|---|---|---|---|---|
| Eosinophils count | ✓ | ✗ | ✗ | ✗ | ✓ |
| Eosinophils percent | ✓ | ✓ | ✗ | ✗ | ✗ |
| Blood urea nitrogen | ✓ | ✗ | ✗ | ✗ | ✓ |
| Ferritin | ✓ | ✗ | ✗ | ✗ | ✗ |
| O2 Saturation | ✓ | ✗ | ✗ | ✗ | ✓ |
| Heart rate | ✓ | ✗ | ✗ | ✗ | ✓ |
| Respiratory rate | ✓ | ✓ | ✗ | ✗ | ✓ |
| O2 Rate | ✓ | ✓ | ✓ | ✓ | ✓ |
| On room air | ✓ | ✓ | ✓ | ✓ | ✓ |
| High O2 support | ✓ | ✓ | ✓ | ✓ | ✓ |
| Age | ✗ | ✗ | ✓ | ✓ | ✗ |

**Table 1: DDLK selects 10/37 features, 8 of which were found to be meaningful by doctors at a large metropolitan hospital.** Here we show the union of `covid-19` features selected by each knockoff method at a nominal FDR of 0.2. Deep Knockoffs, Auto-Encoding Knockoffs, and KnockoffGAN exhibit lower power to select important features.

everywhere else. Auto-Encoding Knockoffs and KnockoffGAN achieve noticeably lower power across all thresholds. Deep Knockoffs perform well here likely due to a lack of strong third or higher moments of dependence between features. We attribute the success of DDLK and Deep Knockoffs to their ability to model highly correlated data.

**COVID-19 adverse events.** [`covid-19`]: The widespread impact of COVID-19 has led to the deployment of machine learning models to guide triage. Data for COVID-19 is messy because of both the large volume of patients and the changing practice for patient care. Establishing trust in models for COVID-19 involves vetting the training data to ensure it does not contain artifacts that models can exploit. Conditional independence tests help achieve this goal in two ways: (a) they highlight which features are most important to the response, and (b) they prune the feature set for a deployed model, reducing the risk of overfitting to processes in a particular hospital. We apply each knockoff method to a large dataset from one of the epicenters of COVID-19 to understand the features most predictive of adverse events.

We use electronic health record data on COVID-positive patients from a large metropolitan health network. Our covariates include demographics, vitals, and lab test results from every complete blood count (CBC) taken for each patient. The response $\mathbf{y} \mid \mathbf{x}$ is a binary label indicating whether or not a patient had an adverse event (intubation, mortality, ICU transfer, hospice discharge, emergency department representation, O2 support in excess of nasal cannula at 6 L/min) within 96 hours of their CBC. There are 17K samples of 37 covariates in the training dataset, 5K in a validation set, and 6K in a held-out test set. We let the DDLK entropy regularization parameter $\lambda = 0.1$. In this experiment, we use gradient boosted regression trees [6, 11] as our $\hat{q}_{\text{response}}(\mathbf{y} \mid \mathbf{x}; \gamma)$ model, and expected log-likelihood as a knockoff statistic. We also standardize the data in the case of Deep Knockoffs since MMDs that use the radial basis function (RBF) kernel with a single bandwidth parameter work better when features are on the same scale.

***Results.*** As COVID-19 is a recently identified disease, there is no ground truth set of important features for this dataset. We therefore use each knockoff method to help discover a set of features at a nominal FDR threshold of 0.2, and validate each feature by manual review with doctors at a

large metropolitan hospital. Table 1 shows a list of features returned by each knockoff method, and indicates whether or not a team of doctors thought the feature should have clinical relevance.

We note that DDLK achieves highest power to select features, identifying 10 features, compared to 5 by DeepKnockoffs, and 4 each by Auto-Encoding Knockoffs and KnockoffGAN. To understand why, we visualize the marginal distributions of each covariate in, and the respective marginal distribution of samples from each knockoff method, in fig. 4.

We notice two main differences between DDLK and the baselines. First, DDLK is able to fit asymmetric distributions better than the baselines. Second, despite the fact that the implementation of DDLK using mixture density networks is misspecified for discrete variables, DDLK is able to model them better than existing baselines. This implementation uses continuous models for $\mathbf{x}$, but is still able approximate discrete distributions well. The components of each mixture appear centered around a discrete value, and have very low variance as shown in fig. 5. This yields a close approximation to the true discrete marginal. We show the marginals of every feature for each knockoff method in appendix G.

## 5  Conclusion

DDLK is a generative model for sampling knockoffs that directly minimizes a KL divergence implied by the knockoff swap property. The optimization for DDLK involves first maximizing the explicit likelihood of the covariates, then minimizing the KL divergence between the joint distribution of covariates and knockoffs and any swap between them. To ensure DDLK satisfies the swap property under any swap indices, we use the Gumbel-Softmax trick to learn swaps that maximize the KL divergence. To generate knockoffs that satisfy the swap property while maintaining high power to select variables, DDLK includes a regularization term that encourages high conditional entropy of the knockoffs given the covariates. We find DDLK to outperform various baselines on several synthetic and real benchmarks including a task involving a large dataset from one of the epicenters of COVID-19.

## Broader Impact

There are several benefits to using flexible methods for identifying conditional independence like DDLK. Practitioners that care about transparency have traditionally eschewed deep learning, but methods like DDLK can present a solution. By performing conditional independence tests with deep networks and by providing guarantees on the false discovery rate, scientists and practitioners can develop more trust in their models. This can lead to greater adoption of flexible models in basic sciences, resulting in new discoveries and better outcomes for the beneficiaries of a deployed model. Conditional independence can also help detect bias from data, by checking if an outcome like length of stay in a hospital was related to a sensitive variable like race or insurance type, even when conditioning on many other factors.

While we believe greater transparency can only be better for society, we note that interpretation methods for machine learning may not exactly provide transparency. These methods visualize only a narrow part of a model's behavior, and may lead to gaps in understanding. Relying solely on these domain-agnostic methods could yield unexpected behavior from the model. As users of machine learning, we must be wary of too quickly dismissing the knowledge of domain experts. No interpretation technique is suitable for all scenarios, and different notions of transparency may be desired in different domains.

## Acknowledgements

Mukund Sudarshan was partially supported by a PhRMA Foundation Predoctoral Fellowship. Mukund Sudarshan and Rajesh Ranganath were partly supported by NIH/NHLBI Award R01HL148248, and by NSF Award 1922658 NRT-HDR: FUTURE Foundations, Translation, and Responsibility for Data Science.

## Footnotes

[2]For all comparison methods, we downloaded the publicly available implementations of the code (if available) and used the appropriate configurations and hyperparameters recommended by the authors. See appendix D for code and model hyperparameters used.

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
