[Supplementary Material]

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

*Proof.* The swap operation $f_H$ on $[\mathbf{x}, \widetilde{\mathbf{x}}]$ swaps coordinates in the following manner: for each $j \in H$, the $j$th and $(j+d)$th coordinates are swapped. Let $(E, \mathcal{E})$ be a measurable space, where elements of $E$ are $2d$-dimensional vectors, and $\mathcal{E}$ is a $\sigma$-algebra on $E$. Let $(F, \mathcal{F})$ also be a measurable space where each element of $F$ is an element of $E$ but with the $j$th coordinate swapped with the $(j+d)$th coordinate for each $j \in H$. Similarly, let $\mathcal{F}$ be constructed by applying the same swap transformations to each element of $\mathcal{E}$. $\mathcal{F}$ is a $\sigma$-algebra as swaps are one-to-one transformations, and $\mathcal{E}$ is a $\sigma$-algebra.

We first show that $f_H$ is a measurable function with respect to $\mathcal{E}$ and $\mathcal{F}$. This is true by construction of the measurable space $(F, \mathcal{F})$. For every element $B \in \mathcal{F}$, $f_H^{-1}(B) \in \mathcal{E}$.

We can now construct a mapping $\mu \circ f_H^{-1}(B)$ for all $B \in \mathcal{F}$. This is the pushforward measure of $\mu$ under transformation $f_H$, and is well defined because $f_H$ is measurable.

Using the fact that a swap applied twice is the identity, we get $f_H = f_H^{-1}$. With this, we see that the probability measure on $(F, \mathcal{F})$ is $\mu \circ f_H^{-1} = \mu \circ f_H$.  $\square$

### A.2  Alternative derivation of theorem 3.1 for continuous random variables

In this section we derive theorem 3.1 for continuous random variables in an alternative manner. Let $\mathbf{x}$ be a set of covariates, and $\widetilde{\mathbf{x}}$ be a set of knockoffs. Let $\mathbf{z} = [\mathbf{x}, \widetilde{\mathbf{x}}]$, and $\mathbf{w} = [\mathbf{x}, \widetilde{\mathbf{x}}]_{\mathrm{swap}(H)}$ where $H$ is a set of coordinates in which we swap $\mathbf{x}$ and $\widetilde{\mathbf{x}}$. Recall that a swap operation on $\mathbf{z}$ is an affine transformation $\mathbf{w} = \mathbf{A}\mathbf{z}$, where $\mathbf{A}$ is a permutation matrix. Using this property, we get:

$$q_{\mathbf{w}}(\mathbf{z}) = \left| \det \left( \frac{\partial \mathbf{A}^{-1}\mathbf{w}}{\partial \mathbf{w}} \right) \right| \cdot q_{\mathbf{z}}(\mathbf{A}^{-1}\mathbf{z}) = q_{\mathbf{z}}(\mathbf{A}^{-1}\mathbf{z}) = q_{\mathbf{z}}(\mathbf{A}\mathbf{z}) = q_{\mathbf{z}}(\mathbf{w}).$$

The first step is achieved by using a change of variables, noting that $\mathbf{A}$ is invertible, and $\mathbf{z} = \mathbf{A}^{-1}\mathbf{w}$. The determinant of the Jacobian here is just the determinant of $\mathbf{A}^{-1}$. $\mathbf{A}^{-1}$ is a permutation matrix whose parity is even, meaning its determinant is 1, and that $\mathbf{A}^{-1} = \mathbf{A}$. I.e. the density of the swapped variables evaluated at $\mathbf{z}$ is equal to the original density evaluated at $\mathbf{w}$.

### A.3  Proof of lemma 3.2

Recall lemma 3.2:

**Worst case swaps**: Let $q(H; \beta)$ be the worst case swap distribution. That is, the distribution over swap indices that maximizes

$$\mathbb{E}_{H \sim q(H;\beta)} \mathrm{KL}(\hat{q}_{\mathrm{joint}}(\mathbf{x}; \theta)\hat{q}_{\mathrm{knockoff}}(\widetilde{\mathbf{x}} \mid \mathbf{x}; \phi) \,\|\, \hat{q}_{\mathrm{joint}}(\mathbf{u}; \theta)\hat{q}_{\mathrm{knockoff}}(\widetilde{\mathbf{u}} \mid \mathbf{u}; \phi))$$

with respect to $\beta$. If this quantity is minimized with respect to $\phi$, knockoffs sampled from $\hat{q}_{\mathrm{knockoff}}$ will satisfy the swap property in eq. (1) for any swap $H$ in the power set of $[d]$.

*Proof.* If

$$\mathbb{E}_{H \sim q(H;\beta)} \mathrm{KL}(\hat{q}_{\mathrm{joint}}(\mathbf{x}; \theta)\hat{q}_{\mathrm{knockoff}}(\widetilde{\mathbf{x}} \mid \mathbf{x}; \phi) \,\|\, \hat{q}_{\mathrm{joint}}(\mathbf{u}; \theta)\hat{q}_{\mathrm{knockoff}}(\widetilde{\mathbf{u}} \mid \mathbf{u}; \phi)) \qquad (9)$$

is minimized with respect to $\phi$ but maximized with respect to $\beta$, then for any other distribution $q(H; \beta')$, eq. (9) will be lesser. Minimizing eq. (9), which is non-negative, with respect to $\phi$ implies that for any swap $H$ sampled from $q(H; \beta)$ and for any knockoff $\widetilde{\mathbf{x}}$ sampled from $\hat{q}_{\mathrm{knockoff}}$,

$$[\mathbf{x}, \widetilde{\mathbf{x}}] \stackrel{d}{=} [\mathbf{x}, \widetilde{\mathbf{x}}]_{\mathrm{swap}(H)}.$$

As eq. (9) is also maximized with respect to $\beta$, swaps $H'$ drawn from all other distributions $q(H'; \beta')$ will only result in lower values of eq. (9). Therefore, the joint distribution $[\mathbf{x}, \widetilde{\mathbf{x}}]$ will be invariant under any swap $H'$ in the power set of $[d]$.  $\square$

### A.4 Sufficient sets for swap condition expectation

Recall the swap property required of knockoffs highlighted in eq. (1):

$$[\mathbf{x}, \widetilde{\mathbf{x}}] \stackrel{d}{=} [\mathbf{x}, \widetilde{\mathbf{x}}]_{\mathrm{swap}(H)}$$

where $H \subseteq [d]$ is a set of coordinates under which we swap the covariates and knockoffs. For valid knockoffs, this equality in distribution must hold for any such $H$. One approach to check if knockoffs are valid is to verify this equality in distribution for all singleton sets $\{j\} \subset [d]$ [20, 10]. To check if the swap property eq. (1) holds under any $H = \{j_1, \ldots, j_k\}$, it suffices to check if eq. (1) holds under each of $\{j_1\}, \ldots, \{j_k\}$.

We can generalize this approach to check the validity of knockoffs under other collections of indices besides singleton sets using the following property. Let $H_1, H_2 \subseteq [d]$ and

$$[\mathbf{x}, \widetilde{\mathbf{x}}] \stackrel{d}{=} [\mathbf{x}, \widetilde{\mathbf{x}}]_{\mathrm{swap}(H_1)}$$

$$[\mathbf{x}, \widetilde{\mathbf{x}}] \stackrel{d}{=} [\mathbf{x}, \widetilde{\mathbf{x}}]_{\mathrm{swap}(H_2)}.$$

Then,

$$[\mathbf{x}, \widetilde{\mathbf{x}}]_{\mathrm{swap}(H_1 \Delta H_2)} \stackrel{d}{=} \left[[\mathbf{x}, \widetilde{\mathbf{x}}]_{\mathrm{swap}(H_1)}\right]_{\mathrm{swap}(H_2)} \stackrel{d}{=} [\mathbf{x}, \widetilde{\mathbf{x}}]_{\mathrm{swap}(H_2)} \stackrel{d}{=} [\mathbf{x}, \widetilde{\mathbf{x}}]$$

where $H_1 \Delta H_2$ is the symmetric difference of $H_1$ and $H_2$. Swapping the indices in $H_1 \Delta H_2$ is equivalent to swapping the indices in $H_1$, then the indices in $H_2$. If $\exists j \in H_1 \wedge j \in H_2$, swapping $j$ twice will negate the effect of the swap.

We can extend this property to $K$ sets and define sufficient conditions to check if the swap property holds. Let $\{A_k\}_{k=1}^{K}$ be a sequence of sets where each $A_k \subseteq [d]$. Let

$$A_1^* = A_1$$

$$\forall k \in [K], A_k^* = A_k \Delta A_{k-1}^*$$

$$A_K^* = \{j\}.$$

Checking the swap property eq. (1) under a sequence of swaps $\{A_k\}_{k=1}^{K}$ is equivalent to checking eq. (1) under the singleton set $\{j\}$. Therefore, the swap property must also hold under the singleton set $\{j\}$.

If collection of sets of swap indices $A$ contains a sub-sequence $\{A_k\}_{k=1}^{K}$ such that their sequential symmetric difference is the singleton $\{j\}$ for each $j \in [d]$, then a set of knockoffs that satisfies the swap property under each $A_k \in A$, will also satisfy the swap property under each singleton set, which is sufficient to generate valid knockoffs.

## B  Implicit reparameterization of mixture density networks

In our experiments, we decompose both $\hat{q}_{\mathrm{joint}}$ and $\hat{q}_{\mathrm{knockoff}}$ via the chain rule:

$$q(\mathbf{x} \mid \cdot) = q(\mathbf{x}_1 \mid \cdot) \prod_{j=2}^{d} q(\mathbf{x}_j \mid \cdot, \mathbf{x}_1, \cdots, \mathbf{x}_{j-1}).$$

We model each conditional $q(\mathbf{x}_j \mid \cdot)$ using mixture density networks [1] which take the form

$$q(\mathbf{x}_j \mid \cdot) = \sum_{k=1}^{K} \pi_k(\cdot; \psi_k) \mathcal{N}(\mu_k(\cdot; \eta_k), \sigma_k^2(\cdot; \omega_k))$$

where functions $\{\pi_k\}_{k=1}^{K}$, $\{\mu_k\}_{k=1}^{K}$, and $\{\sigma_k\}_{k=1}^{K}$ characterize a univariate gaussian mixture. These parameters of these functions are $[\psi_1, \ldots, \psi_K, \nu_1, \ldots, \nu_K, \omega_1, \ldots, \omega_K]$.

**Fitting $\hat{q}_{\mathrm{joint}}$.** Let $\theta$, the parameters of $\hat{q}_{\mathrm{joint}}$ contain parameters for every conditional $q(\mathbf{x}_j \mid \mathbf{x}_1, \ldots, \mathbf{x}_{j-1})$. The optimization of $\theta$ is straightforward:

$$\theta = \arg\max_{\theta} \mathcal{L}(\theta) = \arg\max_{\theta} \frac{1}{N} \sum_{i=1}^{N} \log \hat{q}_{\mathrm{joint}}(\boldsymbol{x}^{(i)}; \theta)$$

only requires taking the derivative of $\mathcal{L}(\theta)$.

**Fitting $\hat{q}_{\textbf{knockoff}}$.** Let $\phi$, the parameters of $\hat{q}_{\text{knockoff}}$ contain parameters for every conditional $q(\widetilde{\mathbf{x}}_j \mid \mathbf{x}_1, \ldots, \mathbf{x}_d, \widetilde{\mathbf{x}}_1, \ldots, \widetilde{\mathbf{x}}_{j-1})$. Recall the loss function $\mathcal{L}(\phi, \beta)$

$$\mathcal{L}(\phi, \beta) = \mathbb{E}_{H \sim \hat{q}_{\text{gumbel}}(\beta)} \mathbb{E}_{\boldsymbol{x} \sim \mathcal{D}_N} \mathbb{E}_{\widetilde{\boldsymbol{x}} \sim \hat{q}_{\text{knockoff}}(\widetilde{\mathbf{x}} \mid \boldsymbol{x}; \phi)} \log \frac{\hat{q}_{\text{joint}}(\boldsymbol{x}; \theta) \hat{q}_{\text{knockoff}}(\widetilde{\boldsymbol{x}} \mid \boldsymbol{x}; \phi)^{1+\lambda}}{\hat{q}_{\text{joint}}(\boldsymbol{u}; \theta) \hat{q}_{\text{knockoff}}(\widetilde{\boldsymbol{u}} \mid \boldsymbol{u}; \phi)}.$$

The optimization of $\phi$ requires $\nabla_\phi \mathcal{L}(\phi, \beta)$, which involves the derivative of an expectation with respect to to $\hat{q}_{\text{knockoff}}(\widetilde{\mathbf{x}} \mid \mathbf{x}; \phi)$. We use implicit reparameterization [5]. The advantage of implicit reparameterization over explicit reparameterization [13] is that an inverse standardization function $S_\phi^{-1}$ – which transforms random noise into samples from a distribution parameterized by $\phi$ – is not needed. Using implicit reparameterization, gradients of some objective $\mathbb{E}_{q(\mathbf{z}; \phi)}[f(\mathbf{z})]$ can be rewritten as

$$\mathbb{E}_{q(\mathbf{z}; \phi)}[\nabla_\phi f(\mathbf{z})] = \mathbb{E}_{q(\mathbf{z}; \phi)}[\nabla_\mathbf{z} f(\mathbf{z}) \nabla_\phi \mathbf{z}]$$
$$= \mathbb{E}_{q(\mathbf{z}; \phi)}[-\nabla_\mathbf{z} f(\mathbf{z})(\nabla_\mathbf{z} S_\phi(\mathbf{z}))^{-1} \nabla_\phi S_\phi(\mathbf{z})].$$

We use this useful property to reparameterize gaussian mixture models. Let $q(\mathbf{z}; \phi)$ be a gaussian mixture model:

$$q(\mathbf{z}; \phi) = \sum_{k=1}^{K} \pi_k \mathcal{N}(\mathbf{z}; \mu_k, \sigma_k^2)$$

where $\phi = [\pi_1, \ldots, \pi_K, \mu_1, \ldots, \mu_K, \sigma_1, \ldots, \sigma_K]$. Let the standardization function $S_\phi$ be the CDF of $q(\mathbf{z}; \phi)$:

$$S_\phi(\mathbf{z}) = \sum_{k=1}^{K} \pi_k \Phi\left(\frac{\mathbf{z} - \mu_k}{\sigma_k}\right)$$

where $\Phi$ is the standard normal gaussian CDF. We use this to compute the gradient of $\mathbf{z}$ with respect to each parameter:

$$\nabla_{\pi_k} \mathbf{z} = -\frac{\Phi\left(\frac{\mathbf{z}-\mu_k}{\sigma_k}\right)}{q(\mathbf{z}; \phi)}$$

$$\nabla_{\mu_k} \mathbf{z} = \frac{\pi_k \cdot \mathcal{N}(\mathbf{z}; \mu_k, \sigma_k^2)}{q(\mathbf{z}; \phi)}$$

$$\nabla_{\sigma_k} \mathbf{z} = \frac{\pi_k \cdot \left(\frac{\mathbf{z}-\mu_k}{\sigma_k}\right) \cdot \mathcal{N}(\mathbf{z}; \mu_k, \sigma_k^2)}{q(\mathbf{z}; \phi)}.$$

Putting it all together, we use the implicit reparameterization trick to implement each conditional distribution in $\hat{q}_{\text{joint}}$ and $\hat{q}_{\text{knockoff}}$.

# C  Implementation details and hyperparameter settings for DDLK experiments

We decompose $\hat{q}_{\text{joint}}$ and $\hat{q}_{\text{knockoff}}$ into the product of univariate conditional distributions using the product rule. We use mixture density networks [1] to parameterize each conditional distribution. Each mixture density network is a 3-layer neural network with 50 parameters in each layer and a residual skip connection from the input to the last layer. Each network outputs the parameters for a univariate gaussian mixture with 5 components. We initialize the network such that the modes are evenly spaced within the support of training data.

Using $\hat{q}_{\text{gumbel}}$, we sample binary swap matrices of the same dimension as the data. As we require discrete samples from the Gumbel-Softmax distribution, we implement a straight-through estimator [9]. The straight-through estimator facilitates sampling discrete indices, but uses a continuous approximation during backpropagation.

The $\hat{q}_{\text{joint}}$ model is optimized using Adam [12], with a learning rate of $5 \times 10^{-4}$ for a maximum of 50 epochs. The $\hat{q}_{\text{knockoff}}$ model is optimized using Adam, with a learning rate of $1 \times 10^{-3}$ for $\phi$ and

$1 \times 10^{-2}$ for $\beta$ for a maximum of 250 epochs. We also implement early stopping using validation loss using the PyTorch Lightning framework [4].

Our code can be found online by installing:

```
pip install -i https://test.pypi.org/simple/ ddlk==0.2
```

**Compute resources.** We run each experiment on a single CPU core using 4GB of memory. Fitting $\hat{q}_{\text{joint}}$ for a 100-dimensional dataset with 2000 samples requires fitting 100 conditional models, and takes roughly 10 minutes. Fitting $\hat{q}_{\text{knockoff}}$ for the same data takes roughly 30 minutes.

Fitting DDLK to our `covid-19` dataset takes roughly 15 minutes in total.

## D    Baseline model settings

For Deep Knockoffs and KnockoffGAN, we use code from each respective repository:

https://github.com/msesia/deepknockoffs
https://bitbucket.org/mvdschaar/mlforhealthlabpub/

and use the recommended hyperparameter settings.

At the time of writing this paper, there was no publicly available implementation for Auto-Encoding Knockoffs. We implemented Auto-Encoding Knockoffs with a VAE with a gaussian posterior $q(\mathbf{z} \mid \mathbf{x}) \approx \mathcal{N}(\mathbf{z}; \mu_{\mathbf{z}}(\mathbf{x}), \sigma_{\mathbf{z}}(\mathbf{x}))$ and likelihood $p(\mathbf{x} \mid \mathbf{z}) \approx \mathcal{N}(\mathbf{x}; \mu_{\mathbf{x}}(\mathbf{z}), \sigma_{\mathbf{x}}(\mathbf{z}))$. Each of $\mu_{\mathbf{z}}, \sigma_{\mathbf{z}}, \mu_{\mathbf{x}}.\sigma_{\mathbf{x}}$ is a 2-layer neural network with 400 units in the first hidden layer, 500 units in the second, and ReLU activations. The outputs of networks $\sigma_{\mathbf{z}}, \sigma_{\mathbf{x}}$ are exponentiated to ensure variances are non-negative. The outputs of network $\mu_{\mathbf{z}}$ and $\sigma_{\mathbf{z}}$ are of dimension $d_{\mathbf{z}}$, and the outputs of $\mu_{\mathbf{x}}$ and $\sigma_{\mathbf{x}}$ are of dimension $d$, the covariate dimension. For each dataset, we choose the dimension $d_{\mathbf{z}}$ of latent variable $\mathbf{z}$ that maximizes the estimate of the ELBO on a validation dataset. In our experiments, we search for $d_{\mathbf{z}}$ over the set $\{d_{\mathbf{z}} : 10 \leq d_{\mathbf{z}} \leq 200, d_{\mathbf{z}} \mod 10 = 0\}$. For each dataset, we use the following $d_{\mathbf{z}}$:

1. `gaussian`: 20
2. `mixture`: 140
3. `gene`: 30
4. `covid-19`: 60.

The neural networks are trained using Adam [12], with a learning rate of $1 \times 10^{-4}$ for a maximum of 150 epochs. To avoid very large gradients, we standardize the data using the mean and standard deviation of the training set. To generate knockoffs $\widetilde{\mathbf{x}}$, we use the same approach prescribed by Liu and Zheng [15]. We first sample the latent variable $\mathbf{z}$ conditioned on the covariates using the posterior distribution:

$$\mathbf{z} \sim \mathcal{N}\left(\mathbf{z}; \mu_{\mathbf{z}}(\mathbf{x}), \sigma_{\mathbf{z}}(\mathbf{x})\right).$$

This sample of $\mathbf{z}$ is then used to sample a knockoff $\widetilde{\mathbf{x}}$ using the likelihood distribution:

$$\widetilde{\mathbf{x}} \sim \mathcal{N}\left(\mathbf{x}; \mu_{\mathbf{x}}(\mathbf{z}), \sigma_{\mathbf{x}}(\mathbf{z})\right).$$

Since these $\widetilde{\mathbf{x}}$ are standardized, we re-scale them by the the training mean and standard deviation.

## E    Robust model-based statistics for FDR-control

The goal of any knockoff method is to help compute test statistics for a conditional independence test. We employ a variant of holdout randomization tests (HRTs) [23] to compute test statistics $w_j$ for each feature $\mathbf{x}_j$. We split dataset $\mathcal{D}_N := \{(\boldsymbol{x}^{(i)}, \boldsymbol{y}^{(i)})\}_{i=1}^N$ into train and test sets $\mathcal{D}_N^{(\text{tr})}$, and $\mathcal{D}_N^{(\text{te})}$ respectively, then sample knockoff datasets $\widetilde{\mathcal{D}}_N^{(\text{tr})}$ and $\widetilde{\mathcal{D}}_N^{(\text{te})}$ conditioned on each. Next, a model $\hat{q}_{\text{response}}$ is fit with $\mathcal{D}_N^{(\text{tr})}$.

To compute knockoff statistics with $\hat{q}_{\text{response}}$, we use a measure of performance $\mathcal{W}(\hat{q}_{\text{response}}, \mathcal{D}_N^{(\text{te})})$ on the test set. For real-valued $\mathbf{y}$, $\mathcal{W}$ is the mean squared-error, and for categorical $\mathbf{y}$, $\mathcal{W}$ is expected log-probability of $\mathbf{y} \mid \mathbf{x}$. A knockoff statistic $w_j := \mathcal{W}(\hat{q}_{\text{response}}, \mathcal{D}_N^{(\text{te})}) - \mathcal{W}(\hat{q}_{\text{response}}, \widetilde{\mathcal{D}}_{j,N}^{(\text{te})})$ is recorded for each feature $\mathbf{x}_j$, where $\widetilde{\mathcal{D}}_{j,N}^{(\text{te})}$ is $\mathcal{D}_N^{(\text{te})}$ but with the $j$th feature swapped with $\widetilde{\mathcal{D}}_N^{(\text{te})}$.

In practice, we use use flexible models like neural networks or boosted trees for $\hat{q}_{\text{response}}$. While the model-based statistic above will satisfy the properties detailed in section 2 and control FDR, its ability to do so is hindered by imperfect knockoffs. In such cases, we observe that knockoff statistics for null features are centered around some $\zeta > 0$, violating a condition required for empirical FDR control. This happens because if the covariates and knockoffs are not equal in distribution, models trained on the covariates will fit the covariates better than the knockoffs and inflate the value of test statistic $w_j$. This can lead to an increase in the false discovery rate as conditionally independent features may be selected if their statistic is larger than the selection threshold. To combat this, we propose a mixture statistic that trades off power for FDR-control.

The mixture statistic involves fitting a $\hat{q}_{\text{response}}$ model for each feature $\mathbf{x}_j$ using an equal mixture of data in $\mathcal{D}_N^{(\text{tr})}$ and $\widetilde{\mathcal{D}}_{j,N}^{(\text{tr})}$, then computing $\mathcal{W}$ as above. Such a $\hat{q}_{\text{response}}$ achieves lower performance on $\mathcal{D}_N^{(\text{te})}$, but higher performance on $\widetilde{\mathcal{D}}_{j,N}^{(\text{te})}$, yielding values of $w_j$ with modes closer to 0, enabling finite sample FDR control. However, this FDR-control comes at the cost of power as the method's ability to identify conditionally dependent features is reduced.

## F    Nonlinear response for `gene` experiments

We simulate the response $\mathbf{y} \mid \mathbf{x}$ for the `gene` experiment using a nonlinear response function designed for genomics settings [14]. The response consists of two first-order terms, a second-order term, and an additional nonlinearity in the form of a `tanh`:

$$k \in [m/4]$$
$$\varphi_k^{(1)}, \varphi_k^{(2)} \sim \mathcal{N}(1,1)$$
$$\varphi_k^{(3)}, \varphi_k^{(4)}, \varphi_k^{(5)}, \varphi_k^{(6)} \sim \mathcal{N}(2,1)$$

$$\mathbf{y} \mid \mathbf{x} = \epsilon + \sum_{k=1}^{m/4} \varphi_k^{(1)} \mathbf{x}_{4k-3} + \varphi_k^{(3)} \mathbf{x}_{4k-2} + \varphi_k^{(4)} \mathbf{x}_{4k-3} \mathbf{x}_{4k-2} + \varphi_k^{(5)} \tanh(\varphi_k^{(2)} \mathbf{x}_{4k-1} + \varphi_k^{(6)} \mathbf{x}_{4k})$$

where $m$ is the number of important features. In our experiments, we set $m = 20$. This means that the first 20 features are important, while the remaining 80 are unimportant.

## G    Generating knockoffs for COVID-19 data

In this section, we visualize the marginals of each feature in our `covid-19` dataset, and the marginals of knockoffs sampled from each method. Figures 5 to 8 plot the marginals of samples from DDLK, KnockoffGAN, Deep Knockoffs, and Auto-Encoding Knockoffs respectively. These provide insight into why DDLK is able to select more features at the same nominal FDR threshold of 0.2. We first notice that knockoff samples from DDLK match the marginals of the data very well. DDLK is the only method that models asymmetric distributions well.

Despite our experiment using mixture density networks to implement DDLK, discrete data is also modeled well. For example the values of `O2_support_above_NC` – a binary feature indicating whether a patient required oxygen support greater than nasal cannula – are also the modes of a mixture density learned by DDLK. Samples from Auto-Encoding Knockoffs, KnockoffGAN, and Deep Knockoffs tend to place mass spread across these values.

**Figure 5:** **The marginal distributions of knockoff samples from DDLK look very similar to those from the data.** Despite this implementation of DDLK using mixture density networks, the modes of each marginal line up with discrete values in the data.

**Figure 6:** The marginals distributions of samples from KnockoffGAN match the data only when the feature $\mathbf{x}_j$ is univariate, and has roughly equal mass on either side of the mode.

**Figure 7:** The marginals distributions of samples from Deep Knockoffs match the data only when the feature $\mathbf{x}_j$ is univariate and has fat tails.

**Figure 8:** Auto-Encoding Knockoffs tend to learn underdispersed distributions for the covariates. Further, all of the marginal distributions learned are univariate and exhibit variance much smaller than that of the data.