[Reviews · NeurIPS 2020]

Review 1

Summary and Contributions: This paper presents a new learning approach for generating knockoffs by relying on deep generative models. While the idea of learning a knockoff sampler using a deep neural network is not new, this paper introduces a novel learning framework as well as an original application for knockoffs. The knockoffs generation problem is formulated as a two-stage learning procedure. In addition, the authors offer a new application to COVID-19 data.

Strengths: Soundness of the claims 1. Theory: the proposed algorithm is motivated by rigorous claims. 2. Empirical evaluation: the methodology seems to perform similarly or even better than existing approaches. Significance and novelty of the contribution 1. The two-stage learning method is novel and includes the following steps: (i) estimation of the distribution of the original features, and (ii) minimization of the KL divergence between the joint distribution of original features and knockoffs, under swapping operations. Also, the authors use the Gumbel-Softmax trick to find swaps that maximize the KL divergence, which is new in the knockoffs sampling literature. Finally, to improve power and avoid trivial solutions, they include a regularization term that encourages the knockoffs to be far from the original features. In sum, this above learning paradigm is novel. Previous knockoffs sampling methods that rely on deep generative models are based on MMD networks or GAN. 2. The paper offers a new real data application. Relevance to the NeurIPS community 1. This work aims at mitigating the ongoing reproducibility crisis in science. 2. The method presents an important application for deep generative models.

Weaknesses: Soundness of the claims The experiments can be further improved as follows: 1. Baseline methods: A comparison with second-order knockoffs (proposed in the original model-X paper) is missing in all experiments. 2. Figures that present the FDR and power as a function of the signal strength are missing in all experiments. See, for example, Figure 3 in the knockoff GAN paper, https://openreview.net/pdf?id=ByeZ5jC5YQ. This is especially relevant for the Gaussian example as the selection of the important variables seems to be easy; the power of all methods is equal to one, even for an FDR level that is equal to zero. 3. Goodness-of-fit tests for synthetic experiments: In Figure 2 the authors investigate the estimation of the marginal distribution, however, it is not clear how well the joint distribution is estimated. Also, this figure is not sufficient as it only visualizes two randomly selected dimensions of the data. The reviewer suggests quantifying the knockoffs quality using the two-sample tests presented in the deep knockoff paper. 4. COVID-19 adverse events: Due to the inherent randomness of the generated knockoffs and fitting algorithms, the reviewer suggests adding to Table 1 the frequencies that correspond to the number of times that each feature is selected by the procedure. This can be implemented by repeating the following two steps many times: (i) resample knockoffs using the trained generator, and (ii) re-run the full variable selection pipeline and record the selected variables. Significance and novelty of the contribution The idea of generating knockoffs using deep generative models is not new. Yet, the approach taken in this paper is novel.

Correctness: While the proposed approach seems to be effective, the empirical methodology can be improved, see "weaknesses" above.

Clarity: Yes

Relation to Prior Work: Previous knockoff samplers rely on the following concepts: (i) a Gaussian approximation of the data, (ii) higher moments approximation using MMD, and (iii) sampling using deep generative adversarial networks. This paper suggests a novel KL-based approach that differs from the above methods, and is interesting.

Reproducibility: Yes

Additional Feedback: Please find below my response to the authors' feedback: 1. Comparison with second-order knockoffs: Thank you for adding such a comparison to the manuscript. 2. FDR and power as a function of the signal strength: The authors explain that their method is robust to the choice of the signal strength (i.e., that the FDR is controlled). However, they did not comment on the trend of power versus signal strength. Therefore, the reviewer still suggests adding FDR/Power vs. signal strength plots to the supplementary material. 3. Measures of knockoff quality and goodness-of-fit tests: The authors observe that the MMD two-sample test has low power and therefore decide not to include a goodness-of-fit test. In response, the reviewer suggests using a classifier two-sample test. Based on the deep knockoffs paper, as well as [REF1], this approach demonstrates better power compared to MMD-based statistics. [REF2] Lopez-Paz, D. and Oquab, M., 2017, April. Revisiting classifier two-sample tests. In International Conference on Learning Representations. https://arxiv.org/pdf/1610.06545.pdf 4. Add feature selection frequencies in Table 1: Thank you for adding this information to the COVID-19 application. To summarize, the paper offers a new way to generate powerful knockoffs. The reviewer believes that addressing points (2) and (3) will further improve the manuscript. I recommend accepting the paper.


Review 2

Summary and Contributions: This paper proposes a direct likelihood method to generate model-X knockoffs. In particular, the method is based on a loss function that quantifies the required pairwise exchangeability for validity and penalizes the conditional entropy of the knockoffs given the original data for power. The loss function is averaged over all possible swaps. Apart from the likelihood formulation and the form of the regularizer, a major difference between this method and previous constructions is on the distribution of the swaps. Instead of using a pre-specified distribution such as the uniform one, the proposed method parametrizes the distribution and optimizes it using the Gumbel-softmax trick. This has the potential to force the algorithm to focus on more adverserial swaps, thereby increasing the validity. In the simulation studies, the proposed method is shown to match the marginal distributions better.

Strengths: I think the Gumbel-softmax trick is very helpful in constructing knockoffs since the set of possible swaps is huge and a pre-specified distribution or sampling scheme is likely missing the unwanted discrepancy. In addition, the paper is well-written.

Weaknesses: I think the paper can be improved if my following concerns are addressed. (1) Roughly speaking, the Gumbel-softmax trick is to approximate a categorical distribution. But in the algorithm, the goal is to sample a subset H. Could you elaborate how the beta's are used to generate H? Moreover, what are the precise definitions of beta's and how is the "temperature" in the softmax trick chosen? (2) In the simulations, the coefficients are set to be 100 / \sqrt{N} * Rademacher(0.5). Does it mean the models are fully dense? Or do you mean "Rademacher" by "Bernoulli" that is supported on {0, 1}? (3) In practical multiple testing problems, the fraction of non-nulls is typically tiny or even zero. It is always better to show a simulation under the global null or with a few non-nulls in which case the error in knockoffs tends to be amplified. (4) To obtain high power, it is desirable to generate knockoffs that are marginally different from the original data. Could you show the distribution of the pairwise correlation between X_j and \tilde{X}_j, or something like the conditional entropy used in your method? This provides a tool of quality control. (5) The estimate of \beta can provide a proxy for the "hardest" swap, or the class of "hardest" swaps. Could you extract them and perform a two-sample test on [x, \tilde{x}] and [x, \tilde{x}]_{swap(H)}, and report the p-value? I think it would be a very interesting scale-free post-hoc diagnostic tool to check the validity of knockoffs. (6) line 104, "that DDLK needs to only model qz , instead of qz". The second "qz" should be "qw". ------------------------ Post-rebuttal update ----------------------- I thank the authors for their responses. The rebuttal clearly addressed most of my concerns so I raised my score to 7. I still felt it is better to include experiments with global nulls. It is true that "Valid knockoffs will yield FDR control regardless of how many important covariates there are". But the knockoffs proposed in this paper is only approximately valid so it is unclear whether it still controls FDR. For models with many non-nulls, all FDR controlling procedures tend to be more robust because once they reject a fair number of non-nulls, they can throw in a few arbitrary rejections without violating inflating FDR much. In the global null case, this unwanted protection goes away. According to my experiences, the global null case is more sensitive in capturing the failure of many "approximately-valid" or "asymptotically-valid" procedures.

Correctness: Yes

Clarity: Yes

Relation to Prior Work: Yes

Reproducibility: Yes

Additional Feedback:


Review 3

Summary and Contributions: Authors present a novel feature knockoff generating scheme that minimizes the KL divergence between the joint distribution of the features with knockoffs and the joint distribution under the worst case possible swap. The authors further provide empirical results on synthetic and semi-synthetic data showing FDR control and outperformance in terms of power. An Interesting application to variable selection for COVID-19 adverse events is also provided.

Strengths: The main contribution of the work is a novel likelihood-based knockoff generating. A minor application contribution that can assist in covid-19 adverse event prediction.

Weaknesses: No major weakness. Perhaps the appendix could have included additional experiments on FDR, and power robustness to synthetic data generation schemes see: Romano, Yaniv, Matteo Sesia, and Emmanuel Candès. "Deep knockoffs." Journal of the American Statistical Association (2019): 1-12.(already in bibliography)

Correctness: Claims and methodology are correct.

Clarity: The manuscript is well written and mostly clear. The results section could be better setout to allow readers to follow clearly.

Relation to Prior Work: Differences with related work are clearly articulated.

Reproducibility: Yes

Additional Feedback: Can the authors please comment on the data source, permission to use and privacy considerations observed in relation to the EHR data used?


Review 4

Summary and Contributions: This paper introduces deep direct likelihood knockoffs, a generative model for knockoffs. The main idea is as follows. The knockoff distribution should be the same as the real distribution; therefore, the KL divergence between these two distributions should be 0. The authors then minimize the KL divergence in order to get matching distributions, where they use the Gumbel-softmax trick. Since this can result in learning a distribution that is correlated with the response, which we don’t want, they impose an entropy regularization. Finally, the authors demonstrate the algorithm's superior performance on several datasets.

Strengths: The idea is good and simple. The theory is pretty straightforward (although not clearly written; see below). Nothing too complicated and I like how the Gumbel-softmax trick was used. Computationally efficient. The experimental results are great!

Weaknesses: Overall, the biggest weakness of this paper is the clarity. Specifically, when writing math it is very important to be precise and thorough because ambiguities are difficult to iron out. Figure 4 is hard to read. The theory in this paper is mostly trivial. This is not a bad thing by itself! In fact, the authors don’t need to prove anything complicated to demonstrate their algorithm, which is great. Complicated, unnecessary math is not useful to anyone. However, I list this as a weakness because the authors demonstrate these results in a way that takes away from the paper. For example, Theorem 1 is correct and the proof is fine, but I wouldn’t call it a theorem. The result, although useful, is almost trivial. The only difficulty in this theorem is knowing that f^{-1} = f. After you know that, you can just apply it to the pushforward mu \circ f^{-1}. For this reason, I would call this a proposition or just state it as a 1-sentence remark. It took me more time to understand Equation 5 than this theorem, which means that the authors’ focus of the mathematical theory takes away from the paper.

Correctness: Line 221 - Some of the comparisons seem bad. For example, the authors mention that KnockoffGAN and Auto-Encoding knockoffs both exhibit mode collapse. Mode collapse is a result of the training failing. This means that the authors compare their model DDLK, which trained successfully, to a model that did not train successfully. This is not really a fair comparison for knockoff generation. Some of the statements that the authors wrote were not clear to me so I mentioned them below. There may be things in the paper that are incorrect and I did not catch them because I thought that the authors meant something else. However, from my understanding most of the paper seems correct.

Clarity: The clarity of this paper is pretty bad. I had to backtrack and reread many times. I had to read between the lines for several mathematical definitions as well. The formatting is weird in several places. Acronyms look like they are in a different font. This is only for some acronyms though. For example, KL is in two different fonts on line 45 and line 50. There is weird spacing on the conditional distribution on line 62. Line 62 - What is y? Is it a number or a vector? This whole paragraph is very confusing. The authors do not define y, x_s, S, [d], w_j, and the distribution in the FDR equation. I am going to assume that the authors mean that [d] = {1,...,d} and that S represents the true subset of useful features. Meaning that some of the features are useless to begin with. We want to estimate S with \hat S. This is not explained in a way that is straightforward and needs to be more clearly presented. Line 39 - The authors say that “Knockoff generation models need to be flexible, so previous methods have focused on likelihood-free generative models, such as generative adversarial networks.” Likelihood-free models do not seem more flexible and it is not clear what is meant by flexibility here. The authors are using this paragraph as motivation and it needs to be more clear. Line 71 - (2) is not an equation. Equation 1 - This seems like equality in distribution but it is not clear. Especially because d is used as a variable to indicate the dimensionality of the data. The authors can leave it, but at least mention in passing that that’s what they are talking about. They do kind of mention it in line 69 but not explicitly. This is important because most stochastic equalities will imply equality in distribution. Line 79 - This paragraph is partly redundant given the paragraph on line 39. Section 2 - This section is mostly based on “CONTROLLING THE FALSE DISCOVERY RATE VIA KNOCKOFFS,” which the authors cite. This is fine, but it is not clear that ALL the material is from that single paper, since the authors simply say “The knockoff filter [1] is a tool used to build variable selection methods” at the beginning. Equation 5 - This one is a bit confusing… Why did we go from q_z and q_x to just q? Are the authors using the equality in distribution from Equation (1) for this? Line 110 - Is this expectation supposed to be an empirical average? Because D_n is not a distribution. Line 141 - Is this expectation over \tilde x? Also, why are there two negatives here?.... Some of the conditions and motivations for the experimental setup are a bit confusing. For example, the way that gene data is selected is quite confusing. It seems that only a subset of genes are selected (100 out of 20,000) so that they are highly correlated. I am not sure why this is done… why do we need correlated features? Maybe this is to make knockoff generation harder but the motivation needs to be clearly stated.

Relation to Prior Work: More relation to previous work would have helped me understand the authors’ original contribution and motivation better.

Reproducibility: Yes

Additional Feedback: If you improve on clarity, this paper will be much, much better.

[Author Response · NeurIPS 2020]

We thank the reviewers for their thoughtful feedback during this difficult time. We are glad they found our writing clear (R2), and our direct likelihood approach to generating knockoffs novel (R1, 3) yet simple (R4). It was particularly encouraging to see the reviewers acknowledge the usefulness of the Gumbel-Softmax trick in ensuring the validity of sampled knockoffs over a large number of potential swaps (R1, 2, 4). We are also glad the application of our method to a timely COVID-19 task stood out (R1, 3). The two types of questions brought up in the reviews involved either (a) possible additional experiments, or (b) clarifications of experimental details or notation. We will answer each question, and have incorporate all feedback into the final version. At the end, we discuss our COVID-19 data source.

*Additional results/experimentation.*   Below, we discuss several thoughtful suggestions brought up by reviewers.

**Comparison with second-order knockoffs** (R1): We compare to Deep Knockoffs, which have been shown to outperform second-order knockoffs (Romano et al. 2018) in a variety of settings. Regardless, we have added a comparison with second-order knockoffs for completeness. **FDR and power as a function of the signal strength** (R1, 3): Both the KnockoffGAN and Deep Knockoffs papers explore this relationship and show that their methods are robust to the choice of signal strength in gaussian data. We observed this trend as well for DDLK, and figured this result would not sufficiently contrast each method. **Measures of knockoff quality and goodness-of-fit tests** (R1, 2): This is a great point and is also realistic for practitioners. We considered using the MMD-based two-sample test from Romano et al. (2018), but found issues as dimensionality increased (even just > 10) making it a poor test for knockoff quality. Ramdas et al. (2015) discuss this issue in detail and suggest that KL is a better measure of discrepancy in high dimensions. However, to evaluate KL, a likelihood is required. At least in the case of DDLK, we can inspect the KL term (L114) on a held-out set. We don't report this statistic in the paper simply because it is not available for likelihood-free models. An extension of our work could include a likelihood-free test for goodness of fit that avoids using MMDs. To check if knockoffs from DDLK will have high power, the conditional entropy term (L141) can be used: higher the conditional entropy, higher the power. We have added a line to the paper discussing empirical checks of knockoff quality. **Add feature selection frequencies in Table 1** (R1): This is a good suggestion, we have added this to Table 1. **Global null** (R2): We felt these experiments would not differ significantly from our existing ones. Valid knockoffs will yield FDR control regardless of how many important covariates there are. Under the global null, the family-wise error rate will also be controlled since in this case FDR = family-wise error rate. **Hardest swaps in two-sample test** (R2): This is a very interesting idea on its own with other potential applications. It definitely seems feasible to create a goodness-of-fit test using the hardest swaps.

*Clarifications of experimental details and notation.*   The clarifications requested are concentrated in one of two categories: details about the Gumbel-Softmax sampling of swaps (R2), and understanding our mathematical notation (R4). We will clarify these below and incorporate all feedback in the final version.

**Sampling swaps with** $\beta$ (R2): For clarity, we quickly recap the full procedure here. The parameter for the $j$th Gumbel-Softmax (GS) is $\beta_j$: the probability of sampling a 1. To sample a swap $H$, we first sample all $d$ GS distributions. If and only if a 1 was sampled from the $j$th GS, then $j \in H$. In practice, the GS is a continuous relaxation of a discrete distribution and will only yield discrete samples if the temperature $\tau \to \infty$. For our implementation of DDLK, we use the straight-through GS estimator (line 174), which allows binary values to be sampled, but uses the continuous approximation in gradient computations. Further, we followed the annealing schedule approach for choosing $\tau$ as prescribed by Jang et al. (2017). We have clarified this process in section 3. **Gaussian experiment response fully dense** (R2): Thank you for pointing this out. The uploaded pdf was missing a sentence that states that 20 out of 100 coordinates were randomly chosen as important. Only these important coefficients get the value $\alpha_j$. The rest get 0. This sentence has been added to the paper. **Second "q" should be "qw"** (R2): Thanks, we corrected this to $q_{\mathbf{w}}$. **Figure 2 visualizes only two dimensions** (R1): Figure 2 is intended to be a diagnostic to help understand differences between each method, rather than an exact measure of goodness-of-fit. **Mode collapse of baseline in L221** (R4): We agree that mode collapse is a sign of unsuccessful training. However, for Deep Knockoffs and KnockoffGAN, we used publicly available implementations. For Auto-Encoding Knockoffs, there was none, so we implemented the method as prescribed in the original paper. Further, we ran an extensive grid search over hyperparameters for each baseline method as discussed in the original papers. We have added a section discussing these in the appendix. **Flexibility of likelihood-free models** (R4): L39 suggests that likelihood-free generative models (e.g. GANs) can be easier to specify for arbitrary covariate distributions because they don't require a specific likelihood. We added a sentence to make this more clear. **Section 2** (R4): We have renamed section 2 to "Background" to remove this confusion. **Correlated gene features** (R4): We chose a dataset where the covariates are highly correlated to make knockoff generation more difficult. We added a sentence to reiterate this fact. **Mathematical notation** (R4): We have added an appendix section explicitly defining each object introduced in section 3 (e.g. what is $\mathbf{y}$, the set $[d]$, etc.).

*COVID-19 data.*   (R3): We are glad the reviewer found the paper interesting and well written. Our COVID-19 EHR data was collected for quality improvement (QI) purposes rather than research, so no IRB approval was required. Through the QI process, the data was de-identified by someone else for another project making our use no longer for human subjects. This data has been made available through the HIPAA Business Associate Agreement to third parties.

[Meta-Review · NeurIPS 2020]

Three referees support acceptance for the contributions, notably a new approach to generating knockoffs based on neural networks that outperforms existing methods for generating knockoffs. R4 dissents, citing the clarity of the paper as is its more significant weakness. I disagree. In fact, this is one of the most clearly written neurips submission in my stack. Nonetheless, I encourage the authors to clarify all the ambiguous mathematical notation R4 flags. R4 also takes issue with the comparison to KnockoffGAN, which experienced mode collapse during training, and hence performed poorly. The author response, however, reveals that the authors used a reference implementation of KnockoffGAN and grid searched for hyperparameters. I commend the authors on their thoroughness here, and recommend acceptance. This submission "checks all the boxes": a novel method that is nonetheless straightforward enough to work in practice, a clear write-up, and compelling experimental results on both synthetic and real data. I encourage the authors to consider revising their manuscript to address R3's point about global nulls.